



# The Arctic picoeukaryote *Micromonas pusilla* benefits from Ocean Acidification under constant and dynamic light

Emily White[1], Clara J.M Hoppe[1], Björn Rost[1, 2]

[1]Alfred-Wegener-Institut – Helmholtz-Zentrum für Polar- und Meeresforschung, Bremerhaven, 27570, Germany

[2]Universität Bremen, FB2, Leobener Strasse, 28359 Bremen, Germany

*Correspondence to: Emily White (ewhite14@msn.com)*

**Abstract.** Compared to the rest of the globe, the Arctic Ocean is affected disproportionately by climate change. Despite
these fast environmental changes, we currently know little about the effects of ocean acidification (OA) on marine key
species in this area. Moreover, the existing studies typically test the effects of OA under constant, hence artificial light fields.
In this study, the abundant Arctic picoeukaryote *Micromonas pusilla* was acclimated to current (400 μatm) and future (1000
μatm) $p\mathrm{CO_2}$ levels under a constant as well as dynamic light, simulating natural light fields as experienced in the upper
mixed layer. To describe and understand the responses to these drivers, growth, particulate organic carbon (POC)
production, elemental composition, photophysiology and reactive oxygen species (ROS) production were analysed. *M.
pusilla* was able to benefit from OA on various scales, ranging from an increase in growth rates to enhanced photosynthetic
capacity, irrespective of the light regime. These beneficial effects were, however, not reflected in the POC production rates,
which can be explained by energy partitioning towards cell division rather than biomass build-up. In the dynamic light
regime, *M. pusilla* was able to optimise its photophysiology for effective light usage during both low and high light periods.
This effective photoacclimation, which was achieved by modifications to photosystem II (PSII), imposed high metabolic
costs leading to a reduction in growth and POC production rates when compared to constant light. There were no significant
interactions observed between dynamic light and OA, indicating that *M. pusilla* was able maintain effective
photoacclimation without increased photoinactivation under high $p\mathrm{CO_2}$. Based on these findings, physiologically plastic *M.
pusilla* may exhibit a robust positive response to future Arctic Ocean conditions.






## 1 Introduction

Alterations to the ecosystem caused by climate change are far more pronounced in the Arctic than in the rest of the
world (Pörtner et al., 2014). The increase in $p\mathrm{CO_2}$ and concomitant decrease in seawater pH, for instance, is particularly fast
in the Arctic Ocean, which is mainly due to the higher solubility of $\mathrm{CO_2}$ at low water temperatures (Bates and Mathis, 2009).
Many studies have investigated the effects of ocean acidification (OA) on phytoplankton and have observed species-specific
responses (Lohbeck et al., 2012; Riebesell and Tortell, 2011; Rost et al., 2008). The negative effects of OA are thought to
result from disturbed ion homeostasis under decreasing pH, while positive responses seem to be driven by the physiological
mechanisms of inorganic carbon uptake (Bach et al., 2013; Rokitta et al., 2012). Photosynthesis requires $\mathrm{CO_2}$ as a substrate
for the carbon-fixing enzyme RubisCO, which is yet characterized by a poor substrate affinity. To avoid $\mathrm{CO_2}$ limitation
arising from this, especially larger phytoplankton typically depend on carbon-concentrating mechanisms (CCMs). These
CCMs involve the transport of $\mathrm{CO_2}$ and/or $\mathrm{HCO_3^-}$ into the cell, the prevention of leakage out of the cell, as well as the
expression of carbonic anhydrase, an enzyme accelerating the interconversion between $\mathrm{CO_2}$ and $\mathrm{HCO_3^-}$ (Reinfelder, 2011)
As CCMs are energetically expensive, a potential downregulation under OA may be beneficial for phytoplankton
(Hopkinson et al., 2011).

Also the rates of warming are two to three times faster than the global average (Trenberth et al., 2007), a
phenomenon known as Arctic amplification (Miller et al., 2010). Warming causes changes to mixing regimes in the surface
ocean (Houghton et al., 2001), probably leading to a shoaling of the mixed layer due to increased thermal stratification and
freshening caused by sea-ice melting (Steinacher et al., 2010). From model predictions, the Arctic Ocean will become nearly
ice-free during the summer more and more frequently (Pachauri et al., 2014). With a decrease in sea-ice cover the primary
productivity in the Arctic is expected to increase due to a longer growing season for phytoplankton (Arrigo et al., 2008). At
the same time, however, annual productivity may get increasingly limited by the low nutrient supply in the Arctic Ocean,
which may decrease even further due to reduced upwelling (Tremblay et al., 2015; Wassmann and Reigstad, 2011). These
environmental changes in the Arctic have already led to changes in community structure and are expected to cause more
dramatic regime shifts in the future (e.g. Nöthig et al., 2015; Li et al., 2009).

Next to climate driven changes, phytoplankton growing in the turbulent upper mixed layer must generally acclimate
changes to the dynamics of light availability. Due to the low light periods in such dynamic light fields, phytoplankton on the
one hand need to increase the light harvesting efficiency, e.g. by increasing the photosynthetic pigments like chlorophyll *a*
(Chl *a*) (Palmer et al., 2013). In the high light periods, on the other hand, photoprotective mechanisms need to be activated to
prevent photodamage to cells (Ragni et al., 2008). There are species-specific differences in the responses to differing light
regimes, which can include changes in the amount of photoprotective pigments, different photorepair mechanisms and
modifications to the number of reaction centres in PSII (Ragni et al., 2008). Such acclimatory responses may be particularly
important in the Arctic shelf seas, where high organic matter loading lead to particularly high light attenuation with depth
(Granskog et al., 2012).

In the world's oceans, the picoplanktonic size fraction (2-3 μm) dominates primary productivity (Worden et al., 2015)
and *Micromonas*-like picoeukaryotes are highly abundant in the Arctic region already today (Lovejoy et al., 2007).
*Micromonas* has been described as a shade-adapted species that can persist in Arctic winter darkness (Marquardt et al.,
2016) with the help of mixotrophy (McKie-Krisberg and Sanders, 2014). Such low-light adapted organisms are thought to
show a lack of plasticity with regard to Chl *a* quota and photoacclimation (Talmy et al., 2013). Smaller phytoplankton are
furthermore expected to particularly benefit from reduced nutrients under enhanced stratification due to their high surface :
volume ratio, which allows them to take up nutrients more efficiently (Brussaard et al., 2013). For the same reasons,
picoeukaryotes may benefit from elevated $p\mathrm{CO_2}$ levels due to increased $\mathrm{CO_2}$ diffusion into the cell. Concurrently, it has been



shown that smaller phytoplankton will thrive under future OA conditions in different experiments (Brussaard et al., 2013; Engel et al., 2008; Hoppe et al., 2017; Hoppe et al., 2018; Maat et al., 2014; Meakin and Wyman, 2011; Schulz et al., 2017)

and are regarded as potential 'winners' of climate change (e.g. Hoppe et al., 2018; Li et al., 2009; Schulz et al., 2017).

Despite their prevalence in all marine habitats, previous studies on OA effects have mostly disregarded the effect of natural light variability. In previous studies, however, fluctuating light has been reported to affect phytoplankton photosynthesis and growth (Falkowski, 1980; Huisman, 1999; Köhler et al., 2018; Litchman, 2000; Litchman et al., 2004). Interactive effects between dynamic light regimes and OA have been observed in the coccolithophore *Gephyrocapsa*

*oceanica*, causing decreased productivity (Jin et al., 2013). Hoppe and co-workers (2015) reported that the Antarctic diatom *Chaetoceros debilis* benefitted from OA under static irradiance, while a dynamic light regime reversed this positive response. This was attributed to the fact that OA-dependent downregulation of the CCM can expose cells to oxidative stress during high light peaks under dynamic light (Hoppe et al., 2015). Oxidative stress occurs when the production of reactive oxygen species (ROS) exceeds the defensive mechanisms for ROS reduction, leading to accumulation in the cells (Apel and

Hirt, 2004). In this study, the response of *M. pusilla* to OA was investigated under constant and dynamic light in order to determine if there was an interactive effect of the two environmental factors. A particular focus was laid on the physiological mechanisms that determined the observed overall responses.

## 90 2 Materials and Methods

### 2.1 Experimental set up

Monoclonal cultures of the picoeukaryote *M. pusilla* (isolated in 2014 by Klara Wolf in Kongsfjorden, Svalbard, 79°N) were
grown in l L glass bottles in semi-continuous dilute batch cultures (max. 158000 cells ml$^{-1}$; diluted every 3-5 days). The temperature remained stable at 2.6 ± 0.2°C. The media was composed of Arctic seawater (from AWI-Hausgarten, 78°N, collected during an RV *Merian* cruise in 2013) filtered through a 0.2 μm membrane filter capsule (Satorius stedium Biotech, Sartobran 300) and enriched with vitamins and trace metals in accordance to the F/2 protocol (Guillard and Ryther, 1962), as well as with macronutrients in Redfield proportions (containing 100 μmol L$^{-1}$ of nitrate and silicate, and 6.2 μmol L$^{-1}$
phosphate).

Both the constant and the dynamic light regime consisted of a 20:4h light:dark cycle with an average light intensity of 83 ± 5 μmol photons m$^2$ s$^{-1}$. The dynamic light regime varying between 0 and 590 μmol photons m$^2$ s$^{-1}$ (Fig. 1). These light levels were calculated based on conditions typically observed on the Arctic Kongsfjorden (Svalbard, 79°N) in late spring, using maximum surface irradiance of 905 μmol photons m$^2$ s$^{-1}$, a mixed-layer depth of 20 m, an extinction coefficient
of 0.35 m$^{-1}$ (C. Hoppe, unpublished results) and a vertical mixing rate of 0.011 m s$^{-1}$ (Denman and Gargett, 1983). Light was supplied through LED lamps (ECONLUX, Solar Stinger Sunstrip, ECONLUX), and the dynamic light regime was regulated using a daylight controller (LED light scaping control, LiWeBe). In both setups, the light levels were monitored using a ULM-500 universal light meter with a 4π sensor (Effeltrich) and light intensity was adjusted with neutral density screens.

The CO$_2$ partial pressures (*p*CO$_2$) were achieved through aeration of the incubation bottles with two different *p*CO$_2$
levels (400 and 1000 μatm) for at least 12 h prior to inoculation. The gas mixtures were created using a gas flow controller (CGM 2000 MCZ Umwelttechnik), which mixed pure CO$_2$ with CO$_2$-free air to the desired *p*CO$_2$ level. The *p*CO$_2$ levels were monitored using a non-dispersive infrared analyser (LI6252; Li Cor Biosciences). The humidified gas mixtures were bubbled through a glass frit and supplied via a sterile 0.2 μm PTFE filter (Midistart 2000, Satorius stedium). Cultures were acclimated to the respective *p*CO$_2$ levels for at least 5 generations prior to the experiment. To minimize shifts in carbonate
chemistry due to biomass production, cell densities were kept low between 5000 and 158000 cells ml$^{-1}$.



## 2.2 Carbonate chemistry

Seawater pH was determined potentiometrically, using a two-point calibrated glass reference electrode (IOline, Schott Instruments) and pH meter (826 pH mobile, Metrohm), and reported on the NBS scale for incubation temperatures. The samples for dissolved inorganic carbon ($C_T$) measurements were gently filtered through a sterile 0.2 µm nalgene syringe filter (Nalgene, Thermo scientific) and stored in in the dark at 2°C in 5 ml borosilicate bottles. The sample was subsequently analysed colorimetrically in duplicate using an auto analyser (Seal Analytical; Stoll et al., 2001) with a reproducibility of ± 8

µmol kg$^{-1}$ (Table 1). A certified reference standard material (CRM) was used to correct for measurement errors (Dickson et al., 2007). The final average $C_T$ values were 2131 ± 17 µmol kg$^{-1}$ at ambient $p$CO$_2$ levels and 2195 ± 12 µmol kg$^{-1}$ under high $p$CO$_2$ (Table 1). The total alkalinity ($A_T$) samples were gently filtered through pre-combusted 25 mm GF/F filters (glass microfiber filter, Whatman, GE Healthcare Life sciences) and stored in 125 ml dark borosilicate bottles at 2°C. Standards and samples were equilibrated to room temperature prior to potentiometric titrations (Brewer et al., 1986) of two 25 ml

subsamples with an auto analyser (Titroline alpha plus, SCHOTT instruments). An internal standard was applied to correct for systematic errors based on measurements of CRMs, and the data was processed using TitriSoft 2.71 software. The corrected final $A_T$ values ranged between 2194 ± 8 µmol kg$^{-1}$ and 2216 ± 6 µmol kg$^{-1}$ (Table 1). The full carbonate system was calculated with a salinity of 32.2 and a temperature of 2°C using the pH and $A_T$ data with the CO$_{2SYS}$ program (Pierrot et al., 2006). The calculations used constants of Mehrbach et al., (1973) with a refit by Dickson and Millero (1987) and a (B)$_T$

value according to Uppström, (1974). The carbonate system remained stable for the duration of the experiment, as $C_T$ drawdown was <2 % and the average daily pH values were 8.13 ± 0.06 for ambient conditions and 7.81 ± 0.03 for high $p$CO$_2$ levels (Table 1).

## 2.3 Growth and cellular composition


Cell densities of *M. pusilla* were quantified using a flow cytometer (FCM; BD Biosciences, Accuri C6). Samples were analysed using live cells, where 490 µl of sample was added to 10 µl of 1 µm microspheres fluorescent beads solution (Polysciences Inc, Fluoresbrite YG), which acted as an internal standard. Cells were identified and counted using the FL3 and FL4 channels as well as forward scatter for 2 minutes on slow speed with a maximum of 50,000 events. Specific growth

rate constants (µ) were calculated from exponential fits of cell numbers over time for each replicate bottle. Samples were measured daily within a 1 h timeframe for consistency.

Samples for particulate organic carbon (POC) and nitrogen (PON) were gently filtered onto pre-combusted 25 mm GF/F filters. Before analysis, 200 µl of hydrochloric acid (HCl, 0.2 µmol L$^{-1}$) was added to each filter and the samples were dried at 60°C for at least 12 h to remove any inorganic carbon. The samples were analysed using an elemental analyser (Euro

EA 3000, HEKAtech). The POC and PON data was corrected by subtracting blank measurements, and values were normalised using the specific cell density and volume filtered to yield cell quotas. Subsequently, production rates were calculated by multiplying the quota with the division rate constant $k$ ($k = \mu/\ln(2)$) of the respective incubation.

Samples for Chl $a$ quota were obtained by gentle filtration onto 25 mm GF/F filters and immediately stored at -20°C until analysis. For chlorophyll extraction, 8 ml of 90% acetone was added to the filters and subsequently stored at 4°C

for at least 2 h. After centrifugation (4500 rpm for 5 min, Sigma 4K10), samples were measured on a fluorometer (TD-700 Fluorometer, Turner designs) before and after acidification with HCl (1 µmol L$^{-1}$). Chl $a$ concentrations (µg L$^{-1}$) were calculated as in Knap et al., (1996).






### 2.4 Physiological responses

Photophysiological parameters were measured using a fast repetition rate fluorometer (FRRf; FastOcean sensor, Chelsea
technologies) in combination with a FastAct system (Chelsea technologies). The fluorometer's light emitting diodes (LEDs)
were set to an emission wavelength of 450 nm. A saturation phase of 100 flashlets on a pitch of 2 µs was used, with a
relaxation phase comprising of 40 flashlets and a pitch of 50 µs. Prior to measurements, samples were dark acclimated for 15
min and measurements were conducted in a temperature-controlled chamber at $3^{\circ}$C. The maximum ($F_m$, $F_m'$) and minimum
($F_o$, $F'$) chlorophyll fluorescence in the dark and light were estimated according to iterative algorithms for induction (Kolber
et al., 1998) and relaxation phase (Oxborough et al., 2012). The PSII quantum yield efficiency was estimated as $F_v/F_m$ using
the following equation:

$$F_v/F_m = (F_m - F_o)/F_m \qquad (1)$$

Additional parameters were measured after dark acclimation, including the absorption cross-section size of PSII
($\sigma_{PSII}$; [$Å^2 \cdot q^{-1}$]), the connectivity of PSII ($\rho$) and the PSII re-opening rate ($\tau$; [ms]), according to Kolber et al., (1998). $F_0'$
was estimated after Oxborough and Baker (1997) as

$$F_0' = F_0 \Big/ \frac{F_v}{F_m} + \frac{F_0}{F_m'} \qquad (2)$$

Thereafter, the coefficient of photochemical quenching qL was calculated after Kramer et al., (2004) as

$$qL = (F_m' - F') / (F_m' - F_0') * (F_0'/F') \qquad (3)$$

The electron transport rates through PSII (ETR; [mol e$^-$ (mol RCII)$^{-1}$s$^{-1}$]) were calculated after Xu et al., (2017)
using the following equation:

$$ETR = \sigma PSII \times qL \times PAR \qquad (4)$$

where $\sigma_{PSII}$ is the absorption cross-section size of PSII, qL is the coefficient of photochemical quenching, and PAR is the
photosynthetically active radiation. Photosynthesis-irradiance (PI) curves were estimated at eight irradiances between 0 and
589 µmol photons m$^{-2}$ s$^{-1}$. According to the model by Webb et al., (1974), the light harvesting efficiency ($\alpha$; [mol e$^-$ m$^2$ (mol
RCII)$^{-1}$ (mol photons)$^{-1}$]) and the maximum relative electron transport rate (ETR$_{max}$; [mol e$^-$ (mol RCII)$^{-1}$ s$^{-1}$]) were estimated
using the following equation:

$$ETR = ETR_{max} \Big[ 1 - e \big( \frac{-\alpha I}{ETR_{max}} \big) \Big] \qquad (5)$$

The light saturation index (I$_k$; [µmol photons m$^{-2}$ s$^{-1}$]) was calculated as ETR$_{max}$ / $\alpha$.

At the maximum light level of 506 µmol photons m$^{-2}$ s$^{-1}$, non-photochemical quenching (NPQ) was calculated as
Y(NPQ) using calculations as described in Klughammer & Schreiber (2008):

$$Y(NPQ) = \frac{F}{F_m'} - \frac{F}{F_m} \qquad (6)$$

Measurements of oxidative stress for both •O$_2^-$ free radicals and H$_2$O$_2$ were assessed using the FCM with the
fluorochromes dihydroethidium (HE; Sigma Aldrich, D7008) and dihydrorhodamine 123 (DHR123; Sigma Aldrich, D1054),
respectively. Methods were adapted from Prado et al., (2012), with final dye concentrations adjusted to 158.5 µM for the
flurochrome HE and 28.87 mM for DHR123, and an optimated incubation time of 30 min in the dark at $2^{\circ}$C. Gated FL1 (505
- 550 nm) and FL3 (600 - 645 nm) detection channels were used to analyse the relative concentration of •O$_2^-$ free radicals
and H$_2$O$_2$, respectively. The oxidative stress measurements were corrected using blank measurements and normalised to cell
size using the forward scatter.

The oxidative stress measurements were taken at two specific time points on the last day of incubation, whereas the
photophysiological measurements were taken solely at time point 2 (Fig. 1). The midday measurements, referred to as time



point 1, were conducted at the highest light intensity (590 μmol photons $m^{-2}$ $s^{-1}$) in the dynamic light cycle, and at the same time under the average light intensity (83 μmol photons $m^{-2}$ $s^{-1}$) in the constant light cycle. The evening measurements, referred to as time point 2, were conducted at the start of the dark period (0 μmol photons $m^{-2}$ $s^{-1}$) in both the constant and dynamic light cycle.


# 3 Results

## 3.1 Growth and cellular composition

In this study, the growth rates of *M. pusilla* were affected by both, light regime and $pCO_2$ level (Fig. 2). Growth was reduced by at least 50 % in dynamic versus constant light, irrespective of $pCO_2$ level (ANOVA, F=562.1, p<0.0001). In addition, growth rates significantly increased (>4 %) under elevated $pCO_2$ levels in both light regimes (ANOVA, F=17.1, p=0.0014).

The POC and PON quotas were not altered by changes in light regime or $pCO_2$ levels (Table 2). The POC production rates were significantly higher in constant versus dynamic light (ANOVA, F=72.6, p<0.0001), irrespective of the $pCO_2$ level applied (Fig. 2). The C:N ratio was not significantly affected by the applied treatments (Fig. 2, Table 2). While Chl *a* quotas decreased significantly under elevated $pCO_2$ levels (ANOVA, F=26.4, p=0.0002), there was no significant response to the light treatments applied (Fig. 2). The applied treatments did not have a significant effect on the C:Chl *a* ratio (Table 2). For

all these parameters, no significant interactive effects between the applied light and $pCO_2$ conditions could be detected.

## 3.2 Photophysiological measurements

The FRRf measurements yielded a number of physiological parameters, most of which were significantly affected by the

different light and/or $pCO_2$ treatments applied (Table 3). The PSII quantum yield efficiency ($F_v/F_m$) under dynamic light was significantly higher compared to the constant light treatment (ANOVA, F=88.5, p<0.0001; Table 3). Even though to a lesser extent, high $pCO_2$ levels also significantly increased the $F_v/F_m$ (ANOVA, F=4.8, p=0.0480; Table 3). The connectivity of PSIIs (ρ) was higher under dynamic versus constant light (ANOVA, F=17.6, p=0.0011), while there was no significant effect of $pCO_2$ (Table 3). Similarly, the absorption cross-section of PSII photochemistry ($\sigma_{PSII}$) was significantly higher in dynamic

compared to constant light (ANOVA, F=77.0, p<0.0001), irrespective of the applied $pCO_2$ level (Table 3). In addition, there was significantly less NPQ under dynamic compared to constant light (ANOVA, F=212.9, p<0.0001), with values being lower under dynamic light (Table 3). However, there was no significant $pCO_2$ response in NPQ (p>0.05). Under dynamic light, the PSII re-opening rate (τ) was significantly and >5 % lower when compared to constant light (ANOVA, F=18.6, p=0.0008), while the τ did not display a significant response to $pCO_2$ (Table 3).

The model by Webb et al., (1974) was used to estimate P-I parameters from the FRRf data. The light saturation index ($I_k$) and maximum photosynthetic rate ($ETR_{max}$) increased both significantly by >10 % under elevated $pCO_2$ levels (ANOVA, F=11.8, p=0.0047 for $I_k$ and F=6.8, p=0.0214 for $ETR_{max}$; Table 3). While there was no significant response of $I_k$ to the two light treatments, $ETR_{max}$ was significantly higher under dynamic light compared to constant light (ANOVA, F=41.2, p<0.0001, Table 3). The light harvesting efficiency (α) was significantly reduced by high $pCO_2$ versus ambient

$pCO_2$ levels (ANOVA, F=9.6, p=0.0084). α was also significantly higher under dynamic light compared to constant light (ANOVA, F=36.0, p<0.0001, Table 3).





### 3.3 Oxidative stress measurements


The relative concentrations of $\cdot O_2^-$ free radicals and $H_2O_2$ were used as an indication of oxidative stress under the applied treatments. At time point 1 (midday), the production of $\cdot O_2^-$ free radicals was not significantly changed in response to $pCO_2$ levels or light regimes (p>0.05; Fig. 3). However, there was a significant increase in $H_2O_2$ production in high $pCO_2$ versus ambient $pCO_2$ conditions (ANOVA, F=4.8, p=0.0488), irrespective of the light treatment applied (Fig. 3). The applied

treatments did not have a significant effect on the oxidative stress measurements at time point 2 (Fig. S1, in the Supplement).

### 4 Discussion

#### 4.1 Effective acclimation towards dynamic light imposes high metabolic costs


In their natural environment, phytoplankton need to cope with varying light in the upper mixed layer (MacIntyre et al., 2000). Next to variation in insolation, the light fields are critically dependent on the mixed layer depth, the light attenuation and the vertical mixing rate. In laboratory experiments, however, they are often exposed to an artificially constant light (Köhler et al., 2018). Simulating a typical light regime in an Arctic Fjord, we could show that *M. pusilla* can photoacclimate

to natural variations in light availability without showing signs of high-light stress. This is supported by significantly higher PSII quantum yield efficiency ($F_v/F_m$) under dynamic light (Table 3), which is commonly used as a health indicator of photosynthetic organisms, indicating successful photoacclimation to varying light intensities (Van Leeuwe and Stefels, 2007).

To achieve this photoacclimation, *M. pusilla* can apparently adjust its PSII physiology to balance photoprotection
during high light periods with sufficient absorption during low-light periods of the dynamic light field. More specifically, there were a number of changes to PSII, including a significant increase in the cross-section size of the antenna in PSII ($\sigma_{PSII}$), an increase in the connectivity between PSIIs ($\rho$) and quicker PSII re-opening rates ($\tau$; Table 3). An increase in $\sigma_{PSII}$ acts to increase the absorption of light (Suggett et al., 2007), which would have been beneficial within the low-light periods of the dynamic light cycle (Schuback et al., 2017), and which is supported by a significant increase in the light harvesting
efficiency under low-light ($\alpha$; Table 3). The observed increase in $\rho$ under dynamic light allows higher flexibility to capture electrons during low light phases while at the same time allowing excess excitation energy to be redistributed among PSII centres during high light phases. This increases energy capture efficiency while protecting the PSII centres from damage through migration of excitation energy between different PSIIs also termed the 'Lake model' (Blankenship, 2014; Trimborn et al., 2014), highlighting that *M. pusilla* has high potential for photoprotection. Additionally, the higher $\tau$ under dynamic
versus constant light indicates more efficient drainage of electrons down-stream of PSII (Kolber et al., 1998). A faster PSII re-opening rate can also compensate for deactivation of functional PSII reaction centres during the high light periods of the dynamic light field (Behrenfeld et al., 1998). The significantly lower NPQ in combination with higher $ETR_{max}$ under dynamic versus constant light reflects photoacclimation to a higher light intensity under dynamic light, allowing effective utilization of high excitation energy without initiating high-light stress (Ragni et al., 2008). Consequently, *M. pusilla* exhibits
the physiological plasticity needed to prevent photodamage, which otherwise can disturb the balance between production and scavenging of reactive oxygen species (ROS), causing oxidative stress and accumulation of ROS (Apel and Hirt, 2004). Indeed, dynamic light did not cause increased ROS production in response to the dynamic light field (Fig. 3). Overall, the PSII physiology of *M. pusilla* was effectively acclimated to naturally varying light, displaying photoprotection strategies during high-light phases and upregulated light harvesting during low-light phases.



The described photoacclimation strategies appear to come at a cost, namely lowered energy transfer efficiency to biomass build-up, which is supported by significantly lower growth rates and POC production, despite an increase in $F_v/F_m$ and $ETR_{max}$ under dynamic compared to constant light (Fig. 2, Table 3). Our findings agree with previous studies, which also found lowered growth under a dynamic light regime (Hoppe et al., 2015; Jin et al., 2013; Mills et al., 2010; Shatwell et al., 2012, Su et al., 2012; Wagner et al., 2006). Changes in light regime strongly influence relationships between

photochemistry, carbon fixation and downstream metabolic processes, optimizing light harvesting to sustain growth (Behrenfeld et al., 2008). In view of this, the significant changes to PSII physiology (Table 3) suggest that resources were channelled towards light harvesting rather than protein synthesis and biomass build-up (Talmy et al., 2013). Therefore, it can be concluded that the lower growth rates in dynamic light were caused by the high metabolic costs associated with photoacclimation to the varying light intensities and not due to photoinhibition. Thus, our results stand in contrast to

previous evidence that suggests *Micromonas* to be a shade adapted genus (Lovejoy et al., 2007), as such low-light adapted species are expected to possess limited plasticity in photoacclimative capabilities (Talmy et al., 2013).

### 4.2 Picoeukaryotes benefit from ocean acidification irrespective of the light regime

The low Arctic temperatures enhance $CO_2$ solubility, and therefore OA, of which photosynthetic organisms may benefit due to increased $CO_2$ availability for photosynthesis (AMAP, 2018). This seems true for picoeukaryotes, as in this study *M. pusilla* showed increased growth rates and photophysiological efficiency under elevated $pCO_2$ (Table 2; Fig. 2). These results are in line with various studies that have reported picoeukaryotes to benefit from increasing $pCO_2$ (Brussaard et al., 2013; Hoppe et al., 2018; Meakin and Wyman, 2011; Newbold et al., 2012; Schaum et al., 2013, Schulz et al., 2017). In the

current study, however, there was no observed increase in POC production (Fig. 2) under higher $pCO_2$ levels, which could be expected assuming lowered costs due to CCM down-regulation (Iglesias-Rodriguez et al., 1998; Rost et al., 2008). The observed increase in growth rates nonetheless indicates beneficial OA effects, potentially due to re-allocation of energy liberated by eased carbon acquisition. Alternatively, the large surface : volume ratio of *M. pusilla* (cell size of 2-3 μm) may generally lower the need for an active CCM, allowing cells to dependent more strongly on diffusive $CO_2$ uptake (Falkowski

and Raven, 2013). As the latter is directly linked to the $pCO_2$ level, it could likewise explain the higher growth rates observed under elevated $pCO_2$. In any case, the growth strategy of the investigated strain involves energy allocation into cell division rather than biomass build-up. Whether picoeukaryotes such as *M. pusilla* benefit from OA due to increased diffusive $CO_2$ uptake or lowered CCM costs remains to be tested.

         To further explain the increase in growth rate under elevated $pCO_2$, it is essential to look into the upstream

physiological parameters. There was a significant increase in the $ETR_{max}$ under OA (Table 3), which indicates an increase in photosynthetic capacity. Previous studies on picoeukaryotes have reported variable results with studies displaying no change or an increase in $ETR_{max}$ in response to OA (Brading et al., 2011; Fu et al., 2007; Kim et al., 2013). In this study, $I_k$ increased in concert with $ETR_{max}$ with increasing $pCO_2$ (Table 3). This '$I_k$-dependent behaviour' is known as acclimation to higher light levels in order to optimize balanced growth (Behrenfeld et al., 2008). In the current case, the increase in $I_k$ under OA

could indicate that eased carbon acquisition shifted the balance of energy acquisition and its sinks towards saturation at higher irradiances, which fits to the reduced Chl *a* quota under these conditions (Fig. 2). At the same time, also the light harvesting efficiency at low light ($\alpha$; please note this unit is per photosystem) decreased in response to OA (Table 3). Such '$I_k$-independent behaviour' is influenced by changed in the relative contribution of different sinks of photosynthetic energy, namely carbon fixation, direct use and ATP generation via cyclic electron transport and other mechanisms (Behrenfeld et al.,

2008). Both photoacclimative strategies mimicked acclimation to high-light in response to increasing $pCO_2$, which may be a general OA response of phytoplankton (e.g. Hoppe et al. 2015, Rokitta & Rost 2012). At the same time, reduced Chl *a* quota indicate that such efficient photosystems decreased the need to invest into the total number of them. This potentially



balances the reductive pressure on the entire cell, as we did not observe any high-light stress, even during peaks (Fig. 3). Although there was a significant increase in $H_2O_2$ concentration under OA relative to ambient $p$CO$_2$ levels, no change in $\cdot O_2^-$ concentration was observed (Fig. 3). Thus, even if ROS production was enhanced under OA, efficient detoxification mechanisms (e.g. reduction of $\cdot O_2^-$ to $H_2O_2$ (Asada, 1999)) seem to be in place. Additionally, changes to $H_2O_2$ concentration have been linked to changes in growth metabolism under non-stressful conditions (Kim et al., 2004), which would fit to the '$I_k$-independent behaviour' observed here, and suggests that sufficient sinks for the enhanced flow of photosynthetic energy were present. Thus, there is ample evidence that, despite no effect on biomass build-up, elevated $p$CO$_2$ facilitated carbon acquisition, led to faster and cheaper photosynthetic energy generation, and higher rates of cell division.

The described changes in photoacclimation were not partnered with a significant increase in POC production, despite an increase in growth rate (Fig. 2). These findings contrast with Hoppe et al., (2018), who reported that POC production rates were generally increased under OA. In this earlier study, however, the Chl $a$ quota of *M. pusilla* remained relatively constant over a large range of $p$CO$_2$ levels at two temperatures, so that OA effects on the ratio between energy allocated into photosynthesis (i.e. Chl $a$) and biomass build-up (i.e. POC) in both studies actually agree. Furthermore, if only the $p$CO$_2$ levels investigated in the current study are considered from Hoppe et al., (2018), varying OA responses (i.e. decreasing vs. increasing for POC production, and constant vs. increasing for Chl $a$ quota) were observed depending on the applied temperatures. This hints towards the well-known fact that even small changes in the environmental conditions can greatly modulate OA-responses of phytoplankton (Riebesell and Gattuso, 2015; Rost et al., 2008). In fact, differences between the two studies could also be caused by differences in the average irradiances (approx. 80 vs. 150 µmol photons m$^{-2}$ s$^{-1}$). Despite these differences one should note, however, that high growth rates were obtained under the various OA treatments. As growth rate is the best available fitness indicator for single strain studies (Schaum and Collins, 2014), our findings is indicative for improved fitness of *M. pusilla* under OA.

### 4.3 *M. pusilla* response does not indicate interactions between light regime and $p$CO$_2$

The interaction between light field and OA has been investigated for the coccolithophore *Gephyrocapsa oceanica* (Jin et al., 2013) and the Antarctic diatom *Chaetoceros debilis* (Hoppe et al., 2015). In both studies, the species' increased their photochemical performance in response to elevated $p$CO$_2$ under constant light. Dynamic light fields reversed the positive effect of high $p$CO$_2$, which was explained by increased high-light stress under OA and a reduction in the energy transfer efficiency from photochemistry to biomass build-up (Hoppe et al., 2015). In the current study, there was no significant interaction between light regime and $p$CO$_2$ (p >0.05, Fig. 2, 3; Table 2, 3). These opposing responses could be caused by group- or species-specific differences in carbon acquisition. Diatoms, for example, have highly effective CCMs (Burkhardt et al., 2001), which are energetically expensive (Hopkinson et al., 2011). As CCMs allow cells to efficiently sink energy under sudden high light (Rost et al., 2006), their downregulation in response to high $p$CO$_2$ can reduce the ability of cells to deal with high-light stress under OA (Hoppe et al., 2015). In contrast to other groups/taxa, which were often found to lose their ability to cope with excess energy under OA and dynamic light (e.g. Gao et al., 2012), *M. pusilla* maintained effective acclimation without photoinactivation under these conditions. This could be attributed to its size, making it less reliant CCMs as a mechanism to reduce reductive pressure under high light, as well as to the observed high plasticity in photophysiological characteristics under dynamic light (Table 3).

In conclusion, the photoacclimation strategies of *M. pusilla* were optimised for the dynamic light field and did not seem to depend on CCMs, therefore the previously observed interaction between $p$CO$_2$ and dynamic light (Gao et al., 2012; Hoppe et al., 2015; Jin et al., 2013) was not observed here. Thus, the results of this study highlight that it is crucial to investigate and understand the underlying physiological mechanisms of observed multi-driver responses in order to judge whether generalizations from individual studies are feasible or not.





### 4.4 Implications for the future Arctic Ocean

The findings of this study highlight the importance of considering a dynamic light field in laboratory studies, as numerous parameters including growth rates were significantly altered when compared to constant irradiance. The common use of constant light fields may substantially underestimate the energetic costs of photoacclimation under in-situ conditions (Köhler et al., 2018). Therefore, dynamic light fields need to be included when predicting future ecosystem functioning. If the responses of the strain used in this study are representative for this species, *M. pusilla* can be expected to cope well with

dynamic light field typical for the surface mixed layer (Table 2, 3). While phytoplankton were often found to suffer from OA under dynamic light (Gao et al., 2012; Hoppe et al., 2015; Jin et al., 2013), *M. pusilla* benefitted slightly from OA irrespective of the light treatment applied. As beneficial effects by OA were also evident under different temperatures (Hoppe et al., 2018), we can conclude that *M. pusilla* has a high plasticity toward OA, warming and difference in light regimes, making it well-adapted for conditions expected for the future Arctic ocean. The high physiological plasticity may

thus also explain why picoeukaryotes are often found to dominate mesocosm assemblages under OA (Brussaard et al., 2013; Engel et al., 2008; Schulz et al., 2017).

Global warming is, due to the phenomenon of Arctic Amplification (Screen and Simmonds, 2010), a particularly important driver for Arctic phytoplankton. *M. pusilla* has been shown to synergistically benefit from OA and warming (Hoppe et al., 2018), but future studies should investigate whether responses differ under dynamic light. Furthermore,

warming causes ocean freshening (Peterson et al., 2002) and enhanced stratification, that further reduces nutrient availability (Steinacher et al., 2010). Picoeukaryotes may also benefit from these anticipated changes in nutrient supply due to their high surface : volume ratio allowing effective nutrient uptake (Li et al., 2009). Changes in the community size structure are biogeochemically important as picoplankton-dominated systems tend to be less efficient with respect to carbon export to depth (Worden et al., 2015). If smaller phytoplankton become more dominant the Arctic pelagic food web, this may benefit

smaller grazers. With this additional steps in the food web, energy transfer efficiency to top-predators as well as into the deep ocean will decrease (Brussaard et al., 2013). Based on the current study, increased abundances of *M. pusilla* under future $p$CO$_2$ levels can be expected not only for the more stable low light environments, but also for the productive mixed layer in spring-time with its dynamic light fields.

**Author contributions**
CJMH designed and supervised the study. EW conducted the research and wrote the paper with contributions from CJMH and BR.

**Competing interest**
The authors declare that they have no conflict of interest.

**Acknowledgements**
We are grateful for field support by the 2014/15 station team of the AWIPEV base in Ny-Ålesund (Svalbard) as well as K. Wolf's help with strain isolation and maintenance of *M. pusilla* cultures. L. Wischnewski, C. Schallenberg, T. Brenneis and
M. Machnik are acknowledged for assistance in the laboratory.



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

**Table 1.** Carbonate chemistry measurements, for each light and $pCO_2$ treatment, from the start and end of the experiment (n=4; mean ± one SD). The measured values are $C_T$ (dissolved inorganic carbon), $A_T$ (total alkalinity) and pH (NBS scale). $pCO_2$ was calculated using the $CO_{2SYS}$ program, with pH and $A_T$ as input values. The values were calculated for 2 °C, with a salinity of 32.2. The nutrient levels were 6.5 µmol kg$^{-1}$ and 100 µmol kg$^{-1}$ for $PO_4$ and $SiOH_4$, respectively.

| Light Treatment | $pCO_2$ [µatm] | Time Point | $C_T$ [µmol kg$^{-1}$] | $A_T$ [µmol kg$^{-1}$] | pH NBS scale | $pCO_2$ [µatm] |
|---|---|---|---|---|---|---|
| Constant Light | 400 | Start | 2105 ± 28 | 2212 ± 6 | 8.19 ± 0.06 | 358 ± 47 |
|  |  | End | 2122 ± 16 | 2194 ± 8 | 8.15 ± 0.07 | 397 ± 64 |
|  | 1000 | Start | 2159 ± 14 | 2216 ± 6 | 7.86 ± 0.01 | 817 ± 24 |
|  |  | End | 2202 ± 6 | 2215 ± 5 | 7.79 ± 0.02 | 956 ± 49 |
| Dynamic Light | 400 | Start | 2134 ± 9 | 2200 ± 6 | 8.11 ± 0.03 | 443 ± 31 |
|  |  | End | 2156 ± 15 | 2207 ± 8 | 8.05 ± 0.03 | 510 ± 37 |
|  | 1000 | Start | 2205 ± 17 | 2204 ± 4 | 7.82 ± 0.03 | 886 ± 53 |
|  |  | End | 2216 ± 11 | 2208 ± 7 | 7.79 ± 0.02 | 963 ± 53 |





**Table 2.** Growth and cellular composition of *M. pusilla* (n=4; mean ± one SD), including the growth rate ($d^{-1}$), the division rate constant ($k$), POC production (fmol cell$^{-1}$ day$^{-1}$), POC quota (fmol cell$^{-1}$), PON quota (fmol cell$^{-1}$), Chlorophyll *a* quota (fg cell$^{-1}$), C:N ratio (mol mol$^{-1}$) and POC : Chl *a* (g g$^{-1}$). Treatments include constant light and dynamic light and the two $p$CO$_2$ levels of 400 µatm and 1000 µatm. The letters indicate significant differences between treatments ($p<0.05$) represented as: (a) light, (b) $p$CO$_2$.

| Light | $p$CO$_2$ | Growth rate $k$ | Division rate constant $\mu$ | POC production | POC quota | PON quota | Chl *a* quota | C:N | POC : Chl *a* |
| --- | --- | --- | --- | --- | --- | --- | --- | --- | --- |
| Treatment | [µatm] | [d$^{-1}$] | [d$^{-1}$] | [fmol cell$^{-1}$ day$^{-1}$] | [fmol cell$^{-1}$] | [fmol cell$^{-1}$] | [fg cell$^{-1}$] | [mol mol$^{-1}$] | [g g$^{-1}$] |
| Constant Light | 400 | 0.70 ± 0.01 | 1.01 ± 0.01 | 179 ± 27 | 177 ± 26 | 19.2 ± 2.3 | 11.5 ± 0.49 | 8.8 ± 0.9 | 185 ± 28 |
| | 1000 | 0.73 ± 0.02 | 1.06 ± 0.03 | 170 ± 31 | 159 ± 29 | 21.2 ± 7.6 | 9.4 ± 0.39 | 9.1 ± 2.4 | 203 ± 36 |
| Dynamic Light | 400 | 0.37 ± 0.01 | 0.53 ± 0.01 | 81 ± 17 | 151 ± 28 | 20.3 ± 4.6 | 10.4 ± 0.39 | 7.7 ± 1.0 | 175 ± 30 |
| | 1000 | 0.41 ± 0.02 | 0.59 ± 0.03 | 77 ± 14 | 132 ± 26 | 18.5 ± 5.7 | 9.3 ± 1.07 | 7.7 ± 1.5 | 172 ± 42 |
| Significance | | a, b | a | a | | | b | | |






**Table 3.** FRRf-based photophysiological parameters for *M. pusilla* (n=4; mean ± one SD). Displayed is the $F_v/F_m$ (dimensionless), the connectivity between PSIIs ($\rho$; [dimensionless]), the absorption cross-section of PSII photochemistry ($\sigma_{PSII}$; [$\text{Å}^2 \cdot q^{-1}$]), the non-photochemical quenching (NPQ; [dimensionless]), the PSII re-opening rate ($\tau$, ms), the maximum photosynthetic rate (ETR$_{max}$; [mol e$^-$ (mol RCII)$^{-1}$ s$^{-1}$]), the light harvesting efficiency ($\alpha$; [mol e$^-$ m$^2$ (mol RCII)$^{-1}$ (mol photons)$^{-1}$]) and the light saturation constant (I$_k$; [µmol photons m$^{-2}$ s$^{-1}$]) for both light regimes and $pCO_2$ levels. The letters indicate significant differences between treatments (p<0.05) represented as: (a) light, (b) $pCO_2$.

| Light | $pCO_2$ [µatm] | $F_v/F_m$ Dimensionless | $\rho$ Dimensionless | $\sigma_{PSII}$ [$\text{Å}^2 \cdot q^{-1}$] | NPQ Dimensionless | $\tau$ [ms] | rETR$_{max}$ [mol e$^-$ (mol RCII)$^{-1}$ s$^{-1}$] | I$_k$ [µmol photons m$^{-2}$ s$^{-1}$] | $\alpha$ [mol e$^-$ m$^2$ (mol RCII)$^{-1}$ (mol photons)$^{-1}$] |
|---|---|---|---|---|---|---|---|---|---|
| Constant Light | 400 | 0.46 ± 0.01 | 0.33 ± 0.08 | 5.2 ± 0.1 | 12.7 ± 1.8 | 617 ± 9 | 369 ± 33 | 61.4 ± 8.0 | 6.0 ± 0.4 |
| | 1000 | 0.49 ± 0.03 | 0.31 ± 0.03 | 5.5 ± 0.1 | 11.4 ± 3.0 | 600 ± 11 | 416 ± 104 | 101.9 ± 33.3 | 4.8 ± 1.1 |
| Dynamic Light | 400 | 0.54 ± 0.00 | 0.40 ± 0.01 | 6.7 ± 0.4 | 0.7 ± 0.0 | 573 ± 24 | 530 ± 40 | 60.1 ± 11.8 | 9.0 ± 1.1 |
| | 1000 | 0.54 ± 0.01 | 0.42 ± 0.02 | 6.8 ± 0.4 | 0.7 ± 0.0 | 569 ± 20 | 640 ± 50 | 84.2 ± 10.4 | 7.6 ± 0.6 |
| Significance | | a, b | a | a | a | a | a, b | b | a, b |







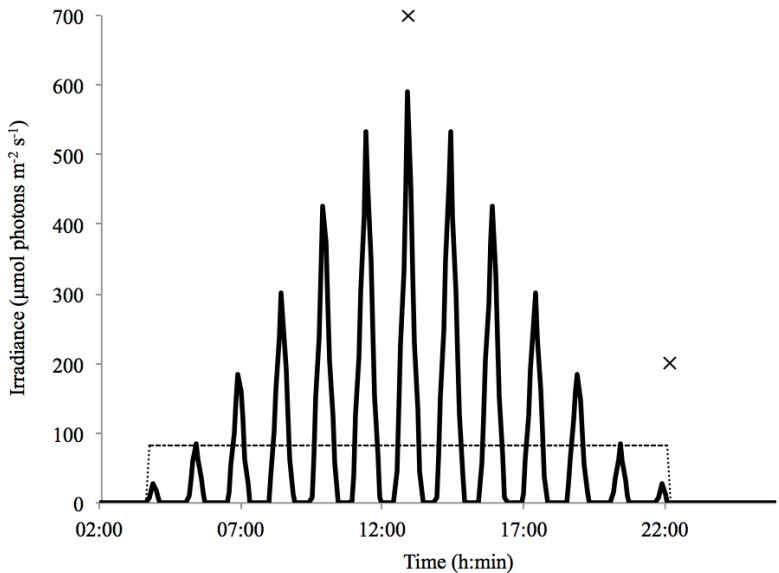

**Figure 1.** Light regimes plotted as a function of time, over a 24 h period. Indicated are the dynamic light cycle (solid line) and the constant light cycle (dashed line). Time point 1 and 2 are displayed at midday, and in the evening at the start of the dark period (x).



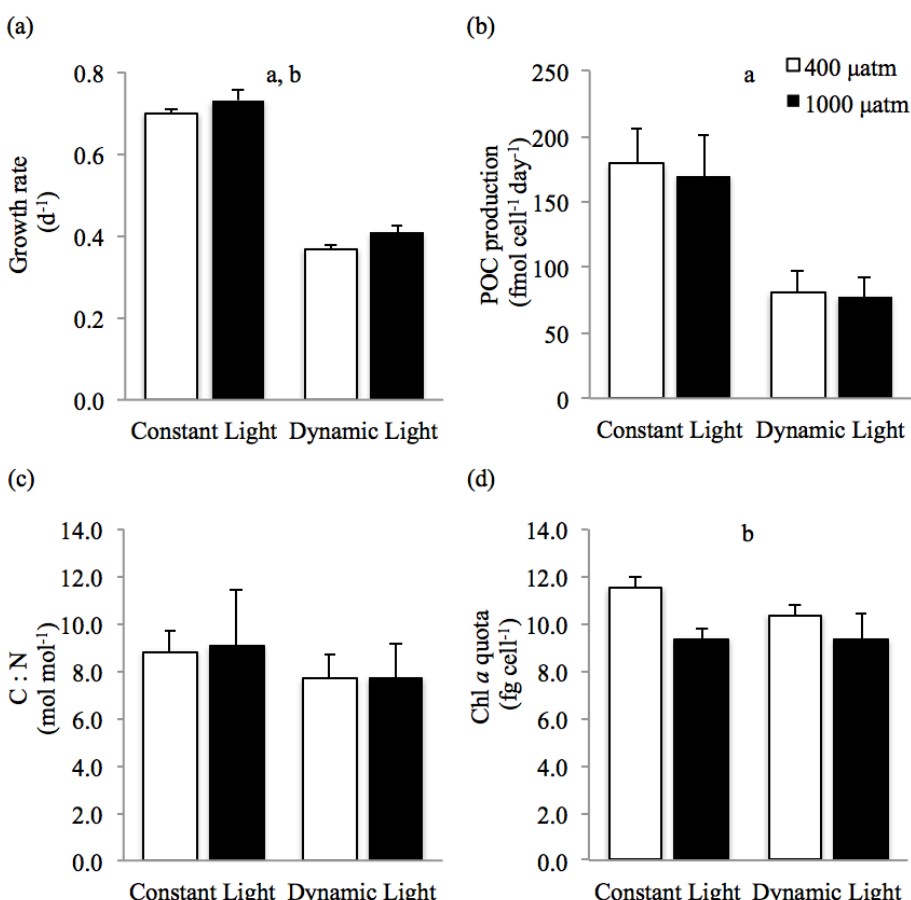


**Figure 2**. **(a)** Growth rate (d[-1]), **(b)** POC production (fmol cell day[-1]), **(c)** C:N ratio (mol mol[-1]), **(d)** Chlorophyll *a* qouta (fg cell[-1]) of *Micromonas pusilla* under constant light and dynamic light and $p\text{CO}_2$ levels of 400 atm (white) and 1000 atm (black;, n=4; mean ± one SD). The letters indicate significant differences between treatments (p<0.05), represented as: (a) Light (b) $p\text{CO}_2$









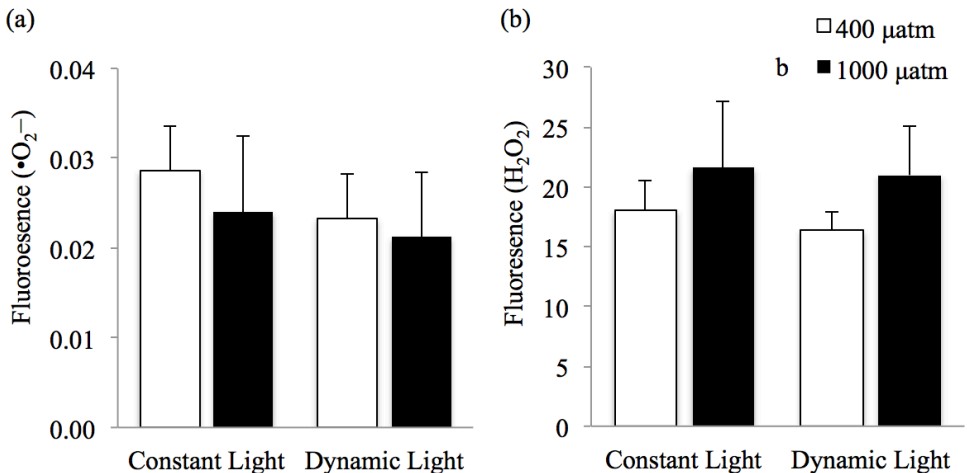

**Figure 3.** The relative production of **(a)** oxygen free radicals ($\cdot O_2-$) and **(b)** hydrogen peroxide ($H_2O_2$) in *Micromonas pusilla* under constant light and dynamic light and $pCO_2$ levels of 400 µatm (white) and 1000 µatm (black; n=4; mean ± one SD) at time point 1. The letters indicate significant differences between treatments (p<0.05), represented as: (b) $pCO_2$



