# Peer review of "S1. Oxidative stress measurements"

_Biogeosciences, 2019_

## Referee Comment (RC1) · Lennart Bach (Referee) · 2 Sep 2019

White et al., investigate the response of Micromonas pusilla to ocean acidification (1000 $\mu$atm) and different types of light supply. The experiments are carefully designed and performed well. This study is very useful as it investigates the response of an important phytoplankton species to OA and different light conditions that has so far not been investigated physiologically in that much detail. I therefore only have minor/moderate comments.

Line 14: Climate change or global warming? I would say the latter although not 100 % sure.

Line 18 (and several times throughout the text): The authors claim that the dynamic light regime resembles a "natural light field". I have doubts that this claim is justified. The underlying assumption is that the organism is repeatedly and regularly moved up and down through the mixed layer. Is this really representative for what is happening in nature? How can chlorophyll a peaks form in the mixed layer if this scenario was true? I am certainly not an expert on this but would assume that the dynamic light regime is also unnatural but differentially unnatural than the constant light. I would therefore suggest rephrasing this claim throughout the text or provide evidence that this scenario is what the cells are typically experiencing.

Line 28: I find the term "physiologically plastic" somewhat cryptic and not necessary in this context.

Line 37ff: This statement is only true for the physiological level. In nature, positive or negative effects can also be induced indirectly e.g. through altered trophic cascades. Please add "on the physiological level" or something like this.

Line 40f: Do you have a reference for this?

Line 65: Two things: 1) According to convention picoplankton is generally considered to be the size class $0.2 - 2$ $\mu$m. 2) I find it hard to believe that this size class "dominates primary productivity" in the oceans. I mean, when diatoms (larger than the picos) already contribute $40 - 50$ % to marine PP than picos would have to contribute the other half (which leaves no room for other important groups such as dinoflagellates). They are without doubt important but I would be very careful with the term "dominant".

Line 120: Was pH measured at incubation temperature or did you correct that somehow after the measurement?

Line 133: Why did you use TA and pH and not TA and DIC? Isn't the DIC measurement more reliable than an NBS based pH measurement.

Line 144: Have you checked if the "slow" flow rate of the Accuri is correct? I calibrated

the flow rates of the Accuri with a balance (measuring sample weight before and after a long measurement test run) and found that the medium and fast flow rates were correct but the slow flow rates were off twofold ( I don't remember out of my head if it was over- or underestimating the flow rate and the data is at the computer at my previous affiliation). I think this is important to clarify as it may significantly alter your growth and production rates, although it probably won't influence your overall interpretation.

Line 148: Please provide molarity of the HCl.

Line 151f: What is the rationale to calculate production rates by multiplying quotas with $\mu/LN(2)$? I am aware that the usual calculation (i.e. quota * $\mu$) is probably not so good but what is the advantage of $\mu$ * k? This very interesting and a sentence explaining this operation would be very helpful.

Lines 264 – 284: Your explanation sounds very plausible but I wonder why is the growth rate so much lower under dynamic light when the cells found a good compromise between low and high light periods. If the cells were as "plastic" as you describe them here, I would expect less of a reduction given that the overall amount of quanta provided to them is the same as in the constant light regime. So, aren't you a bit too optimistic about the performance of the cells or are there examples of other species which "suffered" much more under dynamic light relative to the constant light control setup? In other words, I wonder how you came to the conclusion that M. pusilla was "effectively acclimated" to varying light because I am missing a comparison to a species that is not.

Line 285: This paragraph says that the photoacclimation was costly which makes me wonder if it can be considered "effective" (see previous comment).

Line 294f: Not sure if it is necessary to emphasize this controversy here because there are probably many Micromonas genotypes with different light sensitivities. Prochlorococcus, for example, are known to occur in different water depths (presumably different genotype populations) with different fluorescence signatures. Could also be the case

for Micromonas.

Line 300: The phrasing of "enhancing OA" is kind of weird. Consider rephrasing this sentence.

Line 313: I find this final conclusion a bit too centered on the carbon metabolism. It could also be that the improved "performance" under OA is linked to a pH dependency of nutrient acquisition. We have speculated quite intensely about "why Picoeukaryotes are almost always winning" under simulated OA in natural communities and came up with quite some plausible explanations (I think). Perhaps check out the discussion in this paper (Bach et al., 2017 Plos One, …winners and Loosers in coastal phytoplankton). I am not urging you to cite this paper (!) but just think some speculations therein could enrich this part of the discussion.

Line 317: Do you mean in "our" study? Not sure which study you mean here.

Line 370: I do not understand what kind of generalizations you are talking about. Please clarify.

Line 375: This statement is plausible but how useful is it because you could say exactly the same thing about pretty much every other parameter (trace metal concentration, vitamins temperature etc.) So why is dynamic light a more important variable than any other parameter?

Line 378: see previous comment.

Line 388 and 389: The way you use the term "plasticity" it can basically mean everything. I think it would be better to say precisely what you mean here. Do you mean they benefitted?

Line 389: It seems like the message implied in this sentence is that we would better understand the fate of Micromonas if we combine OA, warming, and dynamic light. I doubt it. Maybe it is not the right place to start this discussion but I think this "multiple stressor" approach without a clear underlying mechanistic framework is leading

nowhere.

Table 1: Are the start and end values mixed up? E.g. DIC is lower at the start than at the end. If this is really the case, why is this so?

Table 2: The calculated pCO2 values are lower than the idealized 400 and 1000 $\mu$atm. Why don't you use measured values for your treatment nomenclature in the text and figures?

Table 2. It is unclear to me what the difference between the growth rate and the division rate constant is. Why do you show both and what does each one mean? The division rate constant is not discussed.

I hope my suggestions help the authors to improve their manuscript.

Kind regards

Lennart Bach

---

## Referee Comment (RC2) · Douglas Campbell (Referee) · 20 Sep 2019

Abstract: Good

Introduction: Good overview of a scattered field.

Figures + legends: Good.

Materials & Methods: "well as with macronutrients in Redfield proportions (containing 100 $\mu$mol L-1 of nitrate and silicate, and 6.2 $\mu$mol L-1 phosphate)."

The goal is to understand what Micromonas might do in a changing Arctic ocean. So how does 100 uM NO3- and 6.2 uM PO4 3- compare to natural levels?

[Figure]

We live in an imperfect world, but responses to an increase in pCO2 (or to fluctuating light) might be very different under a situation of luxury accumulation of excess protein, vs. nutrient limits on protein accumulation etc. Just dimly remembering that 80 uM NO3- is about the equivalent of the Pearl River Delta, so.... pretty high?

I know we face compromises in culture work at getting enough biovolume in a reasonable culture volume, but these points might influence/alter/limit the findings?

In contrast the fluctuating light regime is nicely justified in terms of realistic approximations of the environment.

Eqn. 3 would benefit from an additional set of parentheses around the denominator terms to clarify the order of operations.

Eqn. 4 should use sigmaPSII', otherwise you are not accounting for any non-photochemical down-regulation of sigmaPSII under illumination. If I entered the equation incorrectly in Xu et al. 2017, I apologize, my papers have been filled with equations typos lately.

Line 195: Do these dyes enter cells, or stay outside? or both? I am recently learning that superoxide radical has a very short diffusion length, whereas H2O2 can move a fair ways.

Results: Line 245 The indicator dyes show the standing pool of reactive oxygen, which is the outcome of production rate - detoxification rate. Picky point, but it is possible the effects result from changes in detoxification, rather than production. Also, standing pool of a ROS species is not necessarily the same as oxidative stress...

Lines 285 etc. increased tau under fluctuating light, compared to decreased flow to POC & growth strongly suggests an induction of dissipative electron transport capacity under fluctuating light, leading to 'dumping' of electrons under the high light periods.

Consider that you actually have all the data to estimate the Oxborough proxy for PSII l-1 (based upon F0/sigmaPSII). It is far from perfect, but, if you estimated it, and multiplied by your e- PSII-1 s-1, you could get e- l-1 s-1 Then you can compare electron generation rate with growth rate or with POC accumulation and get an electron quotient for growth. I bet it increases under fluctuating light.

This is perhaps a more defined restatement of your lines 292 etc.

---

## Referee Comment (RC3) · Anonymous Referee #3 · 18 Oct 2019

The manuscript by White et al described the responses of Arctic picoeukaryote Micromonas pusilla to ocean acidification under both constant and dynamic light. The experiments were well designed and performed. The manuscript was well-structured with a good logic flow. However, I do have several minor comments for the revisions before the manuscript be accepted for the publication in BG.

Abstract and Introduction: Good Materials & Methods Line 120: What are the frequencies for the measurements of the pH and did you measure the pH everyday or serval times per day, in the mid-phase of light period or dark period? Please clarify. Line 133: Since the authors measured the carbonate system parameters of pH, TA, and DIC,

why did you calculate the full carbonate system with pH and TA, but not with pH and DIC? Line 147: When did you perform the sampling for POC and PON, at the end of semi-continuous batch culture or in the middle? And when, the middle of light phase or dark phase? Please clarify. The same for Chla. Line 212: What kind of ANOVA did you perform here for the statistical analysis? And I did not see the details about all the statistical analysis that performed in this study. So I would recommend the authors to add a section of "statistical analysis" in the "Materials & Methods" to clarify this issue. And please also report the degree of the Freedom in a standard way for all the stats. Discussion: good.

———————————————————

---

## Author Comment (AC1) · 26 Nov 2019

White et al., investigate the response of Micromonas pusilla to ocean acidification (1000 $\mu$atm) and different types of light supply. The experiments are carefully designed and performed well. This study is very useful as it investigates the response of an important phytoplankton species to OA and different light conditions that has so far not been investigated physiologically in that much detail. I therefore only have minor/moderate comments. We thank the reviewer for his kind words.

Line 14: Climate change or global warming? I would say the latter although not 100 % sure. We would argue that the Arctic is disproportionally affected by climate change in

general, as there have been, for example, strong changes in wind pattern and weather systems (e.g. Hu et al. 2018 Nature Communications, Maturilli & Kayser 2017, Theoretical and Applied Climatology). While warming is the strongest indicator of this trend, ocean acidification is also thought to progress faster than on the global average (Bates & Mathis, 2009, AMAP 2018) it is not the focus of this study. Therefore, we think that it is appropriate to use the term 'climate change' rather than 'global warming' in the abstract. To address the reviewer's point, we have included more detail on this in the introduction, where we now write 'In addition to accelerated rates of OA and warming (Pörtner et al., 2014, Trenberth et al., 2007), the Arctic is affected by strong changes in wind and weather patterns (Hu and Bates, 2018), indicating that this region is disproportionately affected by overall climate change' (L54-56).

Line 18 (and several times throughout the text): The authors claim that the dynamic light regime resembles a "natural light field". I have doubts that this claim is justified. The underlying assumption is that the organism is repeatedly and regularly moved up and down through the mixed layer. Is this really representative for what is happening in nature? How can chlorophyll a peaks form in the mixed layer if this scenario was true? I am certainly not an expert on this but would assume that the dynamic light regime is also unnatural but differentially unnatural than the constant light. I would therefore suggest rephrasing this claim throughout the text or provide evidence that this scenario is what the cells are typically experiencing. We agree with the reviewer that the applied light field is not 'a natural light field' per se, which would be characterized by a lot of stochasticity and higher variability due to turbulent mixing and weather. Nonetheless, the light field we simulate has important characteristics that resemble natural light fields, which are diurnal variations as well as variations as can be expected under simplified homogeneous mixing conditions. As we currently do not have any knowledge on real light fields experiences by cells in the upper mixed layer, we think our approach is nonetheless appropriate. This is especially the case as in the ecosystem where the used M. pusilla strain was isolated from an on which the light field simulation was based on, phytoplankton biomass before and during the spring bloom (i.e.

under non-limiting nutrient conditions) is usually rather evenly distributed in the upper mixed layer (dana not shown). To address the valid criticism on our approach, we now describe in more detail that 'The dynamic light field acts to approximate natural light conditions, assuming homogenous mixing and diurnal changes in incoming irradiance, but no weather-associated variability.' (L107-109), and change our wording throughout the entire manuscript to account for this comment, e.g. 'Simulating a dynamic light field being more representative for an Arctic Fjord' (L272) or 'more realistic variations in light availability' (L273).

Line 28: I find the term "physiologically plastic" somewhat cryptic and not necessary in this context. We agree and deleted these words in the updated manuscript (L27).

Line 37ff: This statement is only true for the physiological level. In nature, positive or negative effects can also be induced indirectly e.g. through altered trophic cascades. Please add "on the physiological level" or something like this. Agreed and done (L36).

Line 40f: Do you have a reference for this? Following the reviewers comment we now have added a reference for this statement (Badger et al. 1998) in L39-40.

Line 65: Two things: 1) According to convention picoplankton is generally considered to be the size class $0.2 - 2$ $\mu$m. 2) I find it hard to believe that this size class "dominates primary productivity" in the oceans. I mean, when diatoms (larger than the picos) already contribute $40 - 50$ % to marine PP than picos would have to contribute the other half (which leaves no room for other important groups such as dinoflagellates). They are without doubt important but I would be very careful with the term "dominant". We thank the reviewer for (1) pointing out the typo regarding the size range of picoplankton. This should read '<$2\mu$m' and will be changed accordingly in the revised manuscript (L66). Regarding (2) their role for global primary production, we now state that the picoplanktonic size fraction (<2 $\mu$m) are significant contributors to overall productivity (L66).

Line 120: Was pH measured at incubation temperature or did you correct that somehow after the measurement? The pH was measured in the same temperature control room as the incubations, but actual temperatures during the measurements differed by (+-1.4°C) due to heat-emitting lamps and insufficient ventilation in the room. For better comparability, the reported pH values have been correct for the specific temperature measured during the pH measurement and the carbonate system has been temperature corrected to 2°C. This procedure is now described in more detail in the methods section of the revised manuscript (L127-128).

Line 133: Why did you use TA and pH and not TA and DIC? Isn't the DIC measurement more reliable than an NBS based pH measurement. According to previous comparisons of an overdetermined carbonate system in our lab (i.e. measuring three instead of two of the parameters and calculating all other from the three possible combinations), the pCO2 calculated from TA and DIC tends to be underestimated by up to 30% (Hoppe et al. 2012). We expect error propagation for measurements with slightly higher uncertainties (i.e. colorimetric DIC measurements and automated small-volume TA titrations instead of large-volume VINDTA measurements) to underlie this systematic error. In the revised manuscript, we now refer to the above-mentioned publication to justify our choices (L140).

Line 144: Have you checked if the "slow" flow rate of the Accuri is correct? I calibrated the flow rates of the Accuri with a balance (measuring sample weight before and after a long measurement test run) and found that the medium and fast flow rates were correct but the slow flow rates were off twofold ( I don't remember out of my head if it was over- or underestimating the flow rate and the data is at the computer at my previous affiliation). I think this is important to clarify as it may significantly alter your growth and production rates, although it probably won't influence your overall interpretation. We thank the reviewer for this very important information regarding potential problem with the Accuri performance. We now checked our data and ran some additional tests with our instrument. All cell count samples were run after adding an internal bead standard solution to account for variability in the flow rate of the FCM (e.g. due to clogging

or variations in pump performance). In n=159 samples, bead counts varied by 1-2 counts $\mu$l-1 only, so that we can assume a very high precision flow rate in all samples. Regarding the accuracy, we have now tested the volumes that the FCM samples in the slow flow settings by weighing the samples before and after the run, aswell as after just inserting the sip without sample collection. The resulting data (n=10) indicates that the instruments specified sample volume has an uncertainty of less than 5%, with the drop that gets stuck on the zip at the end of each measurement accounting for about half of this uncertainty. We therefore conclude that the measurements precision and accuracy are sufficient to use the instruments volumetric data for our calculations.

Line 148: Please provide molarity of the HCl. As written in the manuscript, we used 200 $\mu$l of 0.2 $\mu$mol L-1 HCl. In the revised manuscript, this is written as '0.2 M' (L155).

Line 151f: What is the rationale to calculate production rates by multiplying quotas with_/LN(2)? I am aware that the usual calculation (i.e. quota * _) is probably not so good but what is the advantage of _ * k? This very interesting and a sentence explaining this operation would be very helpful. We decided to calculate production rates as k*POC quota instead of $\mu$*POC quota, as this has confused reviewers in the past because the units of POC production (mol cell-1 day-1) seem to indicate POC production rates per day, but $\mu$ describes e-folding (2.72x) of cell numbers per day, and not doublings in cell numbers per day (and therefore the values of quota and production rate per cell often do not seem to align). After discussing this again and also with a colleague who has been working on such considerations a lot, we now decided to go back to the traditional $\mu$-based calculation (L158-159).

Lines 264 – 284: Your explanation sounds very plausible but I wonder why is the growth rate so much lower under dynamic light when the cells found a good compromise between low and high light periods. If the cells were as "plastic" as you describe them here, I would expect less of a reduction given that the overall amount of quanta provided to them is the same as in the constant light regime. So, aren't you a bit too optimistic about the performance of the cells or are there examples of other species

which "suffered" much more under dynamic light relative to the constant light control setup? In other words, I wonder how you came to the conclusion that M. pusilla was "effectively acclimated" to varying light because I am missing a comparison to a species that is not. We thank the reviewer for this thought stimulating comment. We agree with him that the significant reduction in growth rate (i.e. by 47% under ambient $pCO_2$) questions the effectiveness of the acclimation on the level of biomass build-up. What we actually meant and wrote, however, is that the 'PSII physiology of M. pusilla was effectively acclimated' (L295) to these varying light levels, as can be seem in the lack of any high-light stress indicators and the described adjustments of the photosynthetic machinery to optimize light use during high and low light phases. In our opinion, this does not necessarily need to mean that all quanta are used for biomass buildup in the same way, as cells still spend time under light limitation as well as super-optimal exposure, but merely means that the photosynthetic apparatus does not get damaged under high light despite optimization for low light photon harvest. As we nonetheless understand the reviewer's criticism, we have replaced the words 'effectively acclimated' by 'sufficiently acclimate' (L295). Furthermore, we added a comparison of the changes in growth rate to the next paragraph (see comment below).

Line 285: This paragraph says that the photoacclimation was costly which makes me wonder if it can be considered "effective" (see previous comment). As mentioned above, we were referring to efficient adjustments in PSII physiology and not overall performance. As also expressed in the title of this section, we do make a point that an efficient adjustment of photophysiology to deal with high and low light phases does not mean that the cells actually benefit e.g. in terms of elevated growth rates. We now address this point more explicitly and in greater detail, and compare the changes in growth rates to other published studies by writing 'In previous studies, acclimation to a dynamic light regime has reduced growth rates from 17 % (Hoppe et al., 2015) to 58 % (Boelen et al., 2011), which is comparable to the 47 % reduction in growth rate reported in this study (Table 2; Figure 2). It thus seems likely that such metabolic costs generally occur, and that they are not particularly high in the current study.' (L301-303).
Line 294f: Not sure if it is necessary to emphasize this controversy here because there are probably many Micromonas genotypes with different light sensitivities. Prochloro-coccus, for example, are known to occur in different water depths (presumably different genotype populations) with different fluorescence signatures. Could also be the case for Micromonas. We fully agree with the reviewer's comment. While it is unfortunately beyond the scope of the current study to investigate intra-specific differences within Micromonas pusilla, we accounted for this comment by adjusting the sentence to now read 'our results stand in contrast to previous evidence based on which Micromonas was considered as a generally shade adapted genus (Lovejoy et al. 2007), as such low-light adapted organisms are expected to possess limited plasticity in photoaccli-mative capabilities (Talmy et al., 2013).' (L309-311).

Line 300: The phrasing of "enhancing OA" is kind of weird. Consider rephrasing this sentence. Agreed and done. The sentence now reads 'increase OA' (315).

Line 313: I find this final conclusion a bit too centered on the carbon metabolism. It could also be that the improved "performance" under OA is linked to a pH dependency of nutrient acquisition. We have speculated quite intensely about "why Picoeukaryotes are almost always winning" under simulated OA in natural communities and came up with quite some plausible explanations (I think). Perhaps check out the discussion in this paper (Bach et al., 2017 Plos One, : : :winners and Loosers in coastal phytoplank-ton). I am not urging you to cite this paper (!) but just think some speculations therein could enrich this part of the discussion. We fully agree with the reviewer that it makes a lot of sense to assume that, under natural conditions, the stimulation of picoeukaryote abundances could be linked by indirect effects of OA on nutrient acquisition. Under the here presented experimental conditions, however, inorganic nutrients were provided in such high concentrations, so that we do not expect uptake of organic nutrients or mixotrophy to have a major role in the observed OA responses. We still find the dis-cussion in the mentioned paper relevant and included it in the revised version of the implications chapter (4.4), where we extended the discussion of nutrient limitation, also

in response to a comment of reviewer 2. We now write 'Additionally, nutrient uptake may be facilitated by lower pH under elevated pCO2 (Bach et al., 2017). Nutrient deficiency was not addressed in this study as the experimental design was aiming to mimic non-nutrient limiting conditions before the spring bloom. Nonetheless, the often limiting nutrient supply in the Arctic sets the trophic status of each region and limits annual productivity (Tremblay et al., 2015), thus is an important factor to consider in future studies.' (L411-415).

Line 317: Do you mean in "our" study? Not sure which study you mean here. Agreed and done.

Line 370: I do not understand what kind of generalizations you are talking about. Please clarify. We thank the reviewer for drawing our attention to this lack of clarity. What we wanted to address is the fact that there is a tendency to generalize overarching findings from OA experiments, e.g. 'OA and dynamic light have negative interactive effects on phytoplankton' without considering the underlying physiological mechanisms driving specific responses. If for example, the interaction between OA and dynamic light in diatoms is really driven by the downregulation of CCMs (Hoppe et al. 2015), then one should only expect such an interaction for phytoplankton that strongly rely on CCMs. In the revised manuscript, this respective section now reads 'In conclusion, the photoacclimation strategies of M. pusilla were optimised for the dynamic light field, and as this species seem less dependent on CCMs, the previously described interaction between pCO2 and dynamic light (Gao et al., 2012; Hoppe et al., 2015; Jin et al., 2013) was not observed here. This highlights that, dependent on their various physiological traits, phytoplankton groups may display different types of interactive responses. It is therefore crucial to understand the underlying physiological mechanisms of observed multi-driver responses, in order to judge whether generalizations based on individual studies are feasible or not.' (L381-386).

Line 375: This statement is plausible but how useful is it because you could say exactly the same thing about pretty much every other parameter (trace metal concentration, vitamins temperature etc.) So why is dynamic light a more important variable than any other parameter? While we understand the reviewer's argument here, we think that dynamic light really represents a very important (but largely understudied) component of the environment phytoplankton experience in situ, and one where usual experimental conditions are particularly far from the reality. Most of our mechanistic knowledge of phytoplankton photophysiology has been acquired under constant light fields, and we would argue that the challenges of photoacclimation under highly variable irradiances may have a significant impact on responses to other drivers. This is described more explicitly in the revised manuscript, where we now write 'The findings of this study highlight the importance of considering a dynamic light field in laboratory studies. While the interaction between ocean acidification and other factors, such as higher temperature, can be replicated in the lab (Hoppe et al., 2018), light treatments are generally less representative of in-situ conditions. The difficulty of measuring and simulating more realistic variations in light has meant the common use of constant light fields, which may substantially alter numerous parameters including growth rates and underestimate the energetic costs of photoacclimation under in-situ conditions (Köhler et al., 2018). Therefore, dynamic light fields need to be included when predicting future ecosystem functioning' (L390-395).

Line 378: see previous comment. See above.

Line 388 and 389: The way you use the term "plasticity" it can basically mean everything. I think it would be better to say precisely what you mean here. Do you mean they benefitted? To our knowledge, the term (high) 'physiological plasticity' describes that an organism can deal well with or compensate for a range of different environmental conditions. We agree that this can mean that the organism does benefit from a specific future scenario, but more importantly it can also just mean that the organisms overall performance is not affected too much by different environmental conditions. The latter is the aspect we wanted to highlight here, as such observed physiological acclamatory capacity may allow the organisms to also deal with other scenarios with respect to the

drivers investigated. In the revised manuscript, we are now more explicit and explain that 'physiological plasticity' in this context reefers to the 'the ability to adjust physiologically to maintain high growth and biomass build-up under all tested scenarios' (L401-402).

Line 389: It seems like the message implied in this sentence is that we would better understand the fate of Micromonas if we combine OA, warming, and dynamic light. I doubt it. Maybe it is not the right place to start this discussion but I think this "multiple stressor" approach without a clear underlying mechanistic framework is leading nowhere. We fully agree with the reviewer that such multiple stressor approaches only make sense if they aim at fundamental mechanistic process-understanding. We now specify this is in the manuscript by writing 'future phytoplankton studies should investigate whether responses differ under dynamic light, to determine the mechanisms, metabolic costs and trade-offs associated with interacting physiological processes.' (L407-409).

Table 1: Are the start and end values mixed up? E.g. DIC is lower at the start than at the end. If this is really the case, why is this so? We went back to all the raw data of the carbonate chemistry measurements but did not find an obvious reason (e.g. an obvious swapping of start and end values). Independent pH measurements at an intermediate time point confirmed the final values, we concluded that the data from the initial time point is not trustworthy (likely due to problems with the temperature sensor of the pH meter). We therefore decided to omit the initial values from the table and changed the description of the carbonate chemistry accordingly. Furthermore, we found errors in two single pH values, that were corrected based on the intermediate pH measurements.

Table 2: The calculated pCO2 values are lower than the idealized 400 and 1000 $\mu$atm. Why don't you use measured values for your treatment nomenclature in the text and figures? We generally felt that the target values are better for the reading fluency. Also, due to now using the end and intermediate pH values for the carbonate system calculations (see comment above), the values are now closer to the target values,

therefore we have decided to keep the idealized values.

Table 2. It is unclear to me what the difference between the growth rate and the division rate constant is. Why do you show both and what does each one mean? The division rate constant is not discussed. As discussed above we have decided against using the division rate constant k (i.e. number of doublings per day) for the calculation of the biomass production rates, and now use the growth rate constant $\mu$ instead (i.e. number of e-foldings per day). Therefore, we now also omit the division rate constant k from the revised version of the table.

I hope my suggestions help the authors to improve their manuscript. Kind regards Lennart Bach. We thank the reviewer once again, as we feel his comments really helped to improve our manuscript.

---

## Author Comment (AC2) · 26 Nov 2019

Abstract: Good Introduction: Good overview of a scattered field. Figures + legends: Good. We are happy the reviewer was content with these parts our manuscript.

Materials & Methods: "well as with macronutrients in Redfield proportions (containing 100 _mol L-1 of nitrate and silicate, and 6.2 _mol L-1 phosphate)." The goal is to understand what Micromonas might do in a changing Arctic ocean. So how does 100 uM NO3- and 6.2 uM PO4 3- compare to natural levels? We live in an imperfect world, but responses to an increase in pCO2 (or to fluctuating light) might be very different under a situation of luxury accumulation of excess protein, vs. nutrient limits on protein

accumulation etc. Just dimly remembering that 80 uM NO3- is about the equivalent of the Pearl River Delta, so.... pretty high? I know we face compromises in culture work at getting enough biovolume in a reasonable culture volume, but these points might influence/alter/limit the findings? In contrast the fluctuating light regime is nicely justified in terms of realistic approximations of the environment. We fully agree with the reviewer that low nutrient concentrations are a major feature of the Arctic Ocean. Therefore, it would be highly desirable to conduct experiments under realistically low nutrient levels. It is, however, very tricky to achieve constant levels of nutrient limitation, which would be necessary to fulfil the 'Ceteris paribus' principle and deduce causal relationships. To our knowledge, this is only possible in chemostat or turbidostat set-ups that are very complex to establish and run under low biomass (the latter being necessary for stable carbonate chemistry), and therefore were beyond the scope of this study. Furthermore, the strain we are working with was isolated from pre-spring bloom conditions in a fjord of Svalbard under non-nutrient limiting conditions, so that we are confident that our setup is a reasonable approximation of this habitat/environmental setting. Nonetheless, we now stress the fact that nutrient limitation is a very important driver in the Arctic that we have not investigated in the implications section of our manuscript by writing 'Nutrient deficiency was not addressed in this study as the experimental design was aiming to mimic non-nutrient limiting conditions before the spring bloom. Nonetheless, the often limiting nutrient supply in the Arctic sets the trophic status of each region and limits annual productivity (Tremblay et al., 2015), thus is an important factor to consider in future studies.' (L412-415).

Eqn. 3 would benefit from an additional set of parentheses around the denominator terms to clarify the order of operations. Agreed and done.

Eqn. 4 should use sigmaPSII', otherwise you are not accounting for any nonphotochemical down-regulation of sigmaPSII under illumination. If I entered the equation incorrectly in Xu et al. 2017, I apologize, my papers have been filled with equations typos lately. We thank the reviewer for pointing out this typo. We have used sigmaPSII'

for our calculations, and the equation is corrected in the revised manuscript (L185).

Line 195: Do these dyes enter cells, or stay outside? or both? I am recently learning that superoxide radical has a very short diffusion length, whereas H2O2 can move a fair ways. Both of these dyes enter the cells but also stay in the surrounding seawater (Prado et al. 2012). The HE fluorophore binds DNA after being oxidized by superoxides, while DHR123 localizes in the mitochondria. By using the other flowcytometric characteristics of the cells (forward and side scatter as well as Chla fluorescence), the cell specific levels can be determined and diffusion distances should not play a role. These details are now provided in the text (L199-201).

Results: Line 245 The indicator dyes show the standing pool of reactive oxygen, which is the outcome of production rate - detoxification rate. Picky point, but it is possible the effects result from changes in detoxification, rather than production. Also, standing pool of a ROS species is not necessarily the same as oxidative stress... We fully agree with the reviewer regarding this inaccuracy of our wording, and have replaced 'oxidative stress' by 'ROS' (L255). In the discussion of dynamics light effects, we also slightly changed our wording to clarify such processes, i.e. replacing 'ROS production' by 'ROS accumulation' (L294) while in the discussion of OA effects we already considered these (L345).

Lines 285 etc. increased tau under fluctuating light, compared to decreased flow to POC & growth strongly suggests an induction of dissipative electron transport capacity under fluctuating light, leading to 'dumping' of electrons under the high light periods. Consider that you actually have all the data to estimate the Oxborough proxy for PSII l-1 (based upon F0/sigmaPSII). It is far from perfect, but, if you estimated it, and multiplied by your e- PSII-1 s-1, you could get e- l-1 s-1 Then you can compare electron generation rate with growth rate or with POC accumulation and get an electron quotient for growth. I bet it increases under fluctuating light. This is perhaps a more defined restatement of your lines 292 etc. While there is definitely merit in the calculations of electron transport rates as proposed by Oxborough et al. (2012), we are not confident

that they outweigh the uncertainty introduced by the estimation of PSII concentrations (see e.g. Schuback et al 2016) in this specific dataset. Following the principle of 'as simple as possible, as complex as necessary', we would therefore prefer to keep the simpler e- PSII-1 s-1-based ETR estimate. If the reviewer insists on this point, we would however, be willing to change these calculations.

---

## Author Comment (AC3) · 26 Nov 2019

The manuscript by White et al described the responses of Arctic picoeukaryote Micromonas pusilla to ocean acidification under both constant and dynamic light. The experiments were well designed and performed. The manuscript was well-structured with a good logic flow. However, I do have several minor comments for the revisions before the manuscript be accepted for the publication in BG. We thank the reviewer for their positive comments and will address each of the suggested revisions.

Materials & Methods Line 120: What are the frequencies for the measurements of the pH and did you measure the pH everyday or several times per day, in the mid-phase

of light period or dark period? Please clarify. The pH measurements were conducted at the start, middle and end of the experiment, to check that the carbonate system remained stable throughout the experiment. The start and middle measurement were conducted during the mid-phase of the light period, whereas the end measurement was conducted at the beginning of the dark phase, together with the sampling for all other parameters. For dynamic light this occurred on the first, third and final seventh day of the experiment, however due to faster growth rates under constant light, the measurements occurred on the first, second and fourth day of the experiment. This procedure is now explained in more detail in the revised manuscript (L124-125).

Line 133: Since the authors measured the carbonate system parameters of pH, TA, and DIC, why did you calculate the full carbonate system with pH and TA, but not with pH and DIC? This issue was also brought to our attention by Referee 1, and we provide the following explanation. According to previous comparisons of an overdetermined carbonate system in our lab (i.e. measuring three instead of two of the parameters and calculating all other from the three possible combinations), the pCO2 calculated from TA and DIC tends to be underestimated by up to 30% (Hoppe et al. 2012). We expect error propagation for measurements with slightly higher uncertainties (i.e. colorimetric DIC measurements and automated small-volume TA titrations instead of large-volume VINDTA measurements) to underlie this systematic error. In the revised manuscript, we now refer to the above-mentioned publication to justify our choices (L140).

Line 147: When did you perform the sampling for POC and PON, at the end of semi-continuous batch culture or in the middle? And when, the middle of light phase or dark phase? Please clarify. The same for Chla. The POC, PON and Chl a measurements were all conducted at the end of the experiment, at the beginning of the dark phase. This procedure is now explained in more detail in the revised manuscript (L153 - 154, L160).

Line 212: What kind of ANOVA did you perform here for the statistical analysis? And I did not see the details about all the statistical analysis that performed in this study.

So, I would recommend the authors to add a section of "statistical analysis" in the "Materials & Methods" to clarify this issue. And please also report the degree of the Freedom in a standard way for all the stats. The results were analysed using Mintiab Express statistical software and, a series of Two-way ANOVA tests were performed with a significance level set to p=0.05. We have now added a section on the statistical analysis to the revised manuscript (L212-215). In addition, the statistics in the results section (L224 ff) have been updated to include the degrees of freedom as suggested by the reviewer.
* * *